# STACKED FROM ONE: MULTI-SCALE SELF-INJECTION FOR CONTEXT WINDOW EXTENSION

**Wei Han**[1]*, **Pan Zhou**[2]†, **Shuicheng Yan**[3]
[1] Singapore University of Technology and Design (SUTD), Singapore
[2] Singapore Management University (SMU), Singapore
[3] National University of Singapore (NUS), Singapore

## ABSTRACT

The limited context window of contemporary large language models (LLMs) remains a primary bottleneck for their broader application across diverse domains. Although continual pre-training on long-context data offers a straightforward solution, it incurs prohibitive data acquisition and computational costs. To address this challenge, we propose SHAREDLLM, a novel framework based on multi-grained context compression and query-aware information acquisition. SHAREDLLM comprises two stacked short-context LLMs: a lower model serving as a compressor and an upper model acting as a decoder. The lower model compresses long inputs into compact, multi-grained representations, which are then forwarded to the upper model for context-aware processing. To maximize efficiency, this information transfer occurs exclusively at the lowest layers, bypassing lengthy forward passes and redundant cross-attention operations. This entire process, wherein the upper and lower models are derived from the same underlying LLM layers, is termed *self-injection*. To support this architecture, a specialized tree-based data structure enables the efficient encoding and query-aware retrieval of contextual information. Despite being trained on sequences of only 8K tokens, SHAREDLLM effectively generalizes to inputs exceeding 128K tokens. Across a comprehensive suite of long-context modeling and understanding benchmarks, SHAREDLLM achieves performance superior or comparable to strong baselines, striking an optimal balance between efficiency and accuracy. Furthermore, these design choices allow SHAREDLLM to substantially reduce the memory footprint and yield notable inference speedups ($2\times$ over streaming and $3\times$ over encoder-decoder architectures). The core implementation code, along with training and evaluation details, is open-sourced at `https://github.com/Clement25/SharedLLM`. More detailes are provided in the appendix and supplementary materials.

## 1 INTRODUCTION

Since the release of GPT-3 (Brown, 2020), the rapid advancement of large language models (LLMs) (Chowdhery et al., 2022; Achiam et al., 2023; Touvron et al., 2023a;b; Dubey et al., 2024; Ma et al., 2024; Guo et al., 2025) has revolutionized the NLP research community and transformed various workflows. Pretrained on trillions of tokens, LLMs exhibit remarkable abilities, such as completing unfinished text or code and following human instructions to perform designated tasks after simple supervised fine-tuning (Wei et al., 2021; Chung et al., 2024; Wang et al., 2025). Despite their impressive capabilities, several factors limit their broader application. One major constraint is the *context window* size (Hsieh et al., 2024; Liu et al., 2025), which refers to the maximum number of tokens on which an LLM can behave normally. When the input text exceeds this limit, LLMs may suffer from severe performance degradation or hallucination during inference.

Many researchers have attempted to extend the context window of LLMs with minimal training costs (Peng et al., 2023; Together, 2023; Xiong et al., 2024). One common approach involves post-

---

*Email: wei_han@sutd.edu.sg
†corresponding author: panzhou3@gmail.com

pretraining LLMs on long-context corpora with substantial computational resources (TogetherAI, 2023; Xiong et al., 2024; Ma et al., 2024). Advanced positional encoding methods, which usually extend RoPE to rescale attention scores in a more principled manner, are integrated to minimize the size of training corpus (Chen et al., 2023; Peng et al., 2023). Although they achieve extrapolation— *"train short, test long"*, the efficiency is relatively low. For example, to reach the context length of 128K tokens, using YaRN Peng et al. (2023), one has to pretrain an LLM on 64K tokens. Prompt compression (Ge et al., 2023; Gao et al., 2025) accelerates the inference process by replacing long prompts with LLM-generated semantic tokens, but fails to extend the context window of LLMs or only applies on limited scenarios. Other approaches upgrade the conventional transformer architectures to enable streaming processing of long context (Xiao et al., 2024b; Yen et al., 2024; Zhang et al., 2025), which maintain a sliding window of constant-sized memory. Although these designs significantly alleviate the memory-bound issue of matrix multiplication, their specialized attention patterns may cause incompatibility with high-performance attention implementations (e.g., FlashAttention (Dao et al., 2022; Dao, 2023)), potentially leading to slower inference speeds.

To strike a balance between efficiency and performance, we propose SHAREDLLM, a lightweight architecture which consists of one *upper model* and one *lower model*. The lower model compresses text chunks into multi-grained representations, while the upper model integrates the encoded information and generates the final output. This multi-grained setting helps LLM focus on task-related fine-grained information while regulating other auxiliary coarse-grained information to a secondary role. Both models are initialized from the same *off-the-shelf* checkpoint of a short-context LLM, either in full or in part. Since there is no disparity between the hidden spaces of the two models, SHAREDLLM can be effectively fine-tuned without requiring extensive warmup stages.

This paper makes the following major contributions:

- We propose SHAREDLLM, a hierarchical architecture for efficient LLM context window extension. It consists of two models which work collaboratively through shared key-value mechanism with minimal tunable parameters.

- We design a tree-like structure, called *context tree*, which can express long unstructured context in a coarse-to-fine format. To facilitate this process, we introduces a dynamic context tree construction and search algorithm. Given a context and an query, it can efficiently transform the context into the hierarchical representation and collect relevant information from that tree.

- We conduct a comprehensive experimental study to demonstrate the effectiveness of SHAREDLLM. On the settings of both post-pretraining and supervised fine-tuning, SHAREDLLM shows impressive extrapolation property and yields stronger performance than baseline models with superior memory and time efficiency.

## 2 METHOD

In this section, we first introduce the overall architecture of our proposed SHAREDLLM in Sec. 2.1, and then elaborate on its two main components, the lower model and upper model in Sec. 2.2 and 2.3.

### 2.1 OVERVIEW

As illustrated in Figure 1, SHAREDLLM adopts a hierarchical architecture. The *lower model*, or the "compressor", breaks down the long input context $X_C$ into smaller chunks that are then encoded within limited GPU memory. It then uses the same LLM to compress each context chunk into compact and structured *coarse-to-fine* representations in parallel. The *upper model*, or the "decoder", takes the rear part of the input text (the running context, such as questions) as input. It then integrates the compressed information from the lower model, and finally generates predictions of successive tokens in an auto-regressive manner. The lower and upper models are interconnected via the sharing of key-value (KV) states, which are further integrated at the cross-attention modules in the upper model. To facilitate efficient information gathering and integration, the contextual information processed by the lower model is organized as a binary tree, referred to as the *context tree*, which stores multi-grained information at different levels. Note that the KV compression and transmission occur during the prefilling stage of inference, yet they still improve decoding efficiency because each query

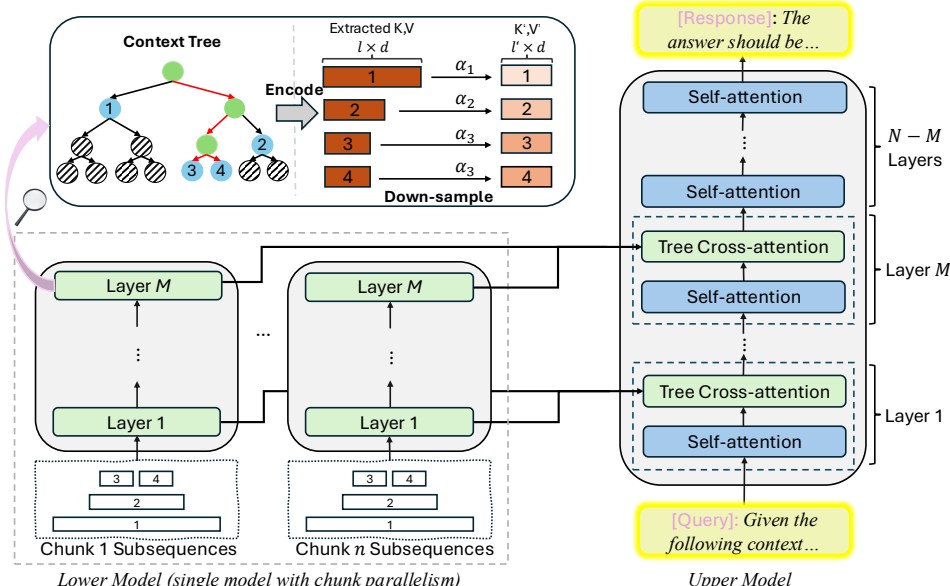

Figure 1: Overview of SHAREDLLM. The architecture resembles general encoder-decoder architecture like T5 (Raffel et al., 2020), but the interaction occurs at the first $M$ layers between lower and upper model through shared key-values which are encoded and compressed from the text chunk into a sequence of trees (top-left).

token attends to a reduced number of key–value pairs. Specifically, this hierarchical decomposition reduces the theoreical self-attention complexity from quadratic $O(T^2)$ to a much more manageable $O(n \cdot (T/n)^2 + T_D \cdot |S'|)$, where $|S'|$ is the highly compressed KV length.

In the following, we elaborate on the lower and upper model. To begin with, we first define some notations to enhance clarity and readability. Let $X = \{x_1, x_2, ..., x_T\}$ represent the entire input sequence, where $T$ denotes the sequence length. We call the LLM whose context window to be extended as "target LLM". Following previous setting (Yen et al., 2024), we split these tokens into two continuous parts: $X = \text{concat}([X_C; X_D])$, where the past context $X_C$ and the running text $X_D$ serve as the inputs to the lower and upper models, respectively. Moreover, the past context $X_C$ is further divided into $n$ smaller and non-overlapping chunks, denoted by $C_1, C_2, ..., C_n$, namely, where $C_1 \cup C_2 \cup ... \cup C_n = X_C$ and $C_i \cap C_j = \emptyset, \forall i \neq j$. The chunk size is controlled to fit within the lower model's context window, allowing the lower model to fully utilize its encoding capacity.

## 2.2  LOWER MODEL

The lower model is a small pretrained LLM, implemented as the first $M$ shallow layers of the target LLM. It independently encodes and compresses each past context chunk $C_i$ from the set of chunks $\{C_i\}_{i=1}^n$, and constructs a context tree that stores multi-grained information across various levels. The encoding process for all chunks $\{C_i\}_{i=1}^n$ is fully paralleled to boost the speed. Below, we detail the context tree structure and its efficiency-enhanced query-dependent dynamic construction, and the tree search process.

**Context Tree.**  The motivation to build the context tree is intuitive and problem-driven. Given a text chunk $C_i$ and a task-specific query, the task-related information is often distributed unevenly across the chunk of text. For instance, to summarize a given passage, one should pay more attention to the topic sentences, collect messages from them and rephrase to produce the answer, rather than focuses much on narrative details. Whereas in the task of passkey finding, detailed relations are more important than theme paragraphs. To this end, we aim for the contextual representations to capture fine-grained details for the relevant portions of the text, while encoding only coarse-grained information for the less relevant parts. The tree structure is the best fit to simulate this process: the

splitting of nodes resembles splitting larger text chunks into smaller ones, from which we can isolate fine-grained information.

The root node of a context tree contains the entire chunk $C_i = \{x_s, ..., x_t\}$, where $x_p$ $(s \leq p \leq t)$ denotes a token, $s$ and $t$ are the start and end index of that chunk; and each other node consists of a sub-sequence of the chunk $C_i$. Then we introduce how to build the child nodes from a parent node. Specifically, for any non-leaf node that contains $l$ tokens $\{x_{u+1}, ..., x_{u+l}\}$, at the training phase, we split it into two sub-sequences to construct its left child and right child as:

$$C_{\texttt{parent}} = \{x_{u+k}\}_{k=1}^{l}, \quad C_{\texttt{left}} = \{x_{u+k}\}_{k=1}^{b}, \quad C_{\texttt{right}} = \{x_{u+k}\}_{k=b+1}^{l}. \tag{1}$$

Here we adopt a random splitting by setting $b = \lfloor \frac{l}{2} - \epsilon \rfloor$ and $\epsilon \sim \mathcal{N}(0, \sigma^2)$ where $\sigma$ is a predefined hyperparameter. This random perturbation serves two critical purposes: first, it acts as a form of structural data augmentation during training, preventing the model from overfitting to fixed splitting positions; second, it mitigates the risk of hard-cutting semantic boundaries (such as splitting in the middle of a crucial entity name) by forcing the model to learn robust representations regardless of exact boundary locations, as concluded in (Zhang et al., 2025). At test time, the noise $\epsilon$ is fixed to zero for deterministic splitting. One can continue this process until arriving at the limited tree depth. Next, building upon this static tree, we construct a more efficient query-dependent dynamic tree.

**Query-Dependent Dynamic Tree Construction and Search.** A task-specific query is typically highly relevant to certain tree nodes while being less relevant to others. For highly relevant nodes, further expansion is necessary to extract fine-grained information. In contrast, for less relevant nodes, expansion is unnecessary. Thus, instead of constructing an entire static context tree as aforementioned, we build a query-dependent dynamic tree that expands only the relevant nodes, as shown in Figure 2, significantly saving both GPU memory and time.

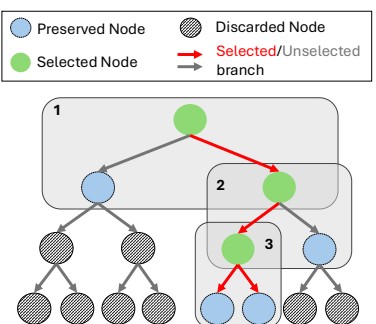

Starting from the root node, we perform a depth-first splitting and search process. Each node sequence is first divided into two subsequences according to Eq. (1). We then use a non-parametric policy $\pi$ to decide the next selected node based on the two subsequences, $\vec{x}_{\texttt{left}}$ and $\vec{x}_{\texttt{right}}$, and a query sequence $\vec{y}$:

Figure 2: An running example of our tree (depth=3). Each box indexed by $i$ represents the $i$th iteration of node split and selection.

$$\pi((\vec{x}_{\texttt{left}}, \vec{x}_{\texttt{right}}), \vec{y}) \rightarrow \texttt{left} \text{ or } \texttt{right}, \tag{2}$$

Here the policy $\pi$ determines whether the left or right child of the node will be selected. The unselected sibling node is marked as "preserved" and will not be expanded further. Note, the root node is always selected to ensure expansion. For policy $\pi$, it is task-specific. Specifically, for language modeling tasks (where the LLM behaves like the non-SFT model), we keep selecting the right branch to simulate the useful $\Lambda$-shape pattern (Han et al., 2024; Ge et al., 2024):

$$\pi((\vec{x}_{\texttt{left}}, \vec{x}_{\texttt{right}}), \vec{y}) \equiv \texttt{right}. \tag{3}$$

For instruction-following tasks (where the LLM serves as the supervised finetuned version), where queries are explicit and available, $\pi$ selects the node with higher semantic similarity to the query:

$$\pi((\vec{x}_{\texttt{left}}, \vec{x}_{\texttt{right}}), \vec{y}) = \underset{\phi \in \{\texttt{left}, \texttt{right}\}}{\arg\max} (\mathbf{sim}(\vec{h}_{\vec{x}_\phi}, \vec{h}_{\vec{y}})), \tag{4}$$

where $\mathbf{sim}(\cdot, \cdot)$ represents the cosine similarity of two vectors. The hidden vector $\vec{h}$ at the last position of a sequence is embedded by either the lower or upper model. Specifically, this involves a short forward pass through one self-attention layer in the lower model for $\vec{h}_{\vec{x}_\phi}$ and the upper model for $\vec{h}_{\vec{y}}$. Once the selected node is determined, the process continues with that node, repeating the procedure until reaching leaf nodes. At this point, both the left and right child are marked as "preserved".

For each preserved node, we feed its associated context into the lower model to obtain a collection of key-value (KV) states from all $M$ layers, denoted as $\mathbf{S} = \{\mathbf{K}, \mathbf{V}\}$, where $\mathbf{K}, \mathbf{V} \in \mathbb{R}^{M \times l \times d}$

represents the key and value states for all $M$ layers. Here, $l$ is the sequence length, and $d$ is the hidden dimension. Next, we perform a uniform downsampling along the length dimension to retain only a portion of the KV states, resulting in $\mathbf{S}' = \{\mathbf{K}', \mathbf{V}'\}$, where $\mathbf{K}', \mathbf{V}' \in \mathbb{R}^{M \times l' \times d}$ and $l'$ are the downsampled length. We implement this uniform downsampling via fractional strided selection to ensure equidistant preservation of context tokens without introducing additional parametric pooling layers. The compression ratio $\alpha$ for the node is defined as $\alpha = l/l'$. For the context tree, we apply a constant compression ratio $\alpha_w$ for all preserved nodes at level $w$, but the ratio diminishes progressively from top to bottom, i.e., $\alpha_w > \alpha_{w+1}$. In our implementation, we set $\alpha_w = 2\alpha_{w+1}$. This approach creates *coarse-to-fine* distribution of semantic information from top to down: nodes at higher levels possess longer subsequences and are compressed with a higher compression ratio, corresponding to more coarse-grained information, while on the contrary, nodes closer to the bottom store fine-grained information. The overall compression ratio $\beta$ of a tree is defined as the ratio of the chunk length $|C|$ to the total length of the compressed KV states:

$$\beta = \frac{\sum l_w n_w}{\sum l'_w n_w} = \frac{|C|}{\sum l'_w n_w} \tag{5}$$

where $n_w$ is the number of preserved nodes at level $w$, and $l'_w$ is the compressed length of each preserved node at level $w$. For the convenience of parallel processing, we set $\beta$ to be the same value for all $n$ context trees. Experimental results in Section 3 demonstrate that this compression ratio can reach as high as 8, significantly improving efficiency.

## 2.3 UPPER MODEL

The upper model shares a similar architecture with the full-layer version of the base model, except for the inserted cross-attention layers that interact with the lower model, as illustrated in Figure 1.

**Position-aware Cross-attention on the Context Tree.** In Section 2.2, we can obtain a sequence of tree-structural representations $\mathcal{S}' = \{\mathbf{S}'_1, ..., \mathbf{S}'_n\}$ for $n$ chunks $\{C_i\}_{i=1}^n$, where $\mathbf{S}'_i = \{\mathbf{K}'_i, \mathbf{V}'_i\}$ stands for the representations of chunk $C_i$. Since the sequence of chunk keys $\mathcal{K} = \{\mathbf{K}'_1; ...; \mathbf{K}'_n\}$ is produced from ordered chunks $\{C_1, ..., C_n\}$, their positional information should be aware at the chunk level by the query to maintain the global chronological order of the original text. To this end, we assign the following chunk-level positional indices to $\mathbf{Q}$ and $\mathcal{K}$:

$$\mathbf{P_Q} = \{\underbrace{n, n, ..., n}_{|X_D|}\}, \quad \mathbf{P_K} = \{\underbrace{0, 0, ..., 0}_{|C_1|/\beta}, \underbrace{1, 1, ..., 1}_{|C_2|/\beta}, \underbrace{n-1, n-1, ..., n-1}_{|C_n|/\beta}\}. \tag{6}$$

Here we view the upper model's query $\mathbf{Q}$ as one chunk and endow it with the largest positional index because $\mathbf{Q}$ is encoded from $X_D$ which is behind all context chunks $X_C$ in the raw input sequence $X$. We then apply rotary positional embedding (RoPE) to $\mathbf{Q}$ and $\mathcal{K}$ according to these block indices. By doing so, the cross-attention mechanism naturally respects the relative distances between the query and each compressed chunk.

In the cross-attention layer, we calculate attention results between the query $\mathbf{Q}$ and concatenated KVs to integrate their carried context information into the running context for more coherent language modeling. To ensure smooth integration with the upper model's original self-attention states, the cross-attention output is fused residually:

$$O = \mathrm{cross\_attn}(\mathbf{Q}, \mathtt{concat}([\mathbf{K}'_1; ...; \mathbf{K}'_n]), \mathtt{concat}([\mathbf{V}'_1; ...; \mathbf{V}'_n])). \tag{7}$$

**Training** We use the standard language modeling loss during training, which maximizes the log probability of the ground-truth tokens in the target sequences $X_{\mathrm{tar}}$, conditioned on the context $X_C$ and all preceding tokens $x_{<t}$ from $X_D$:

$$\mathcal{L} = - \sum_{x_t \in X_{\mathrm{tar}}} \log P(x_t | X_C; x_{<t}).$$

For language modeling data, $X_{\mathrm{tar}} = X_D$, i.e., the target tokens are all tokens in $X_D$, excluding the first token. For instruction-following data, $X_D$ includes both the instruction $X_{\mathrm{inst}}$ and the annotated response $X_{\mathrm{res}}$. In this case, we set $X_{\mathrm{tar}} = X_{\mathrm{res}}$, meaning that we optimize only for the response tokens, while the instruction text is masked during loss calculation.

## 3 EXPERIMENT

### 3.1 SETUP

We highlight some key experimental settings in this section. For more detailed information, please refer to Section A.1.

**Dataset** For language modeling, we follow Yen et al. (2024) to prepare the training data by sampling a subset of 20B (1%) tokens from RedPajama (Together, 2023). Due to the copyright issue, the books3 subset is no longer available and thus excluded from our training set. We will give an analysis towards the impact by this in Section A.4. The sampled texts are truncated to 8,192 tokens for training. In SFT, we follow Zhang et al. (2025) to prepare the dataset. More details can be found in the appendix.

**Training** We initialize the upper model with short-context LLMs, such as LLaMA-2-7B, LLaMA-3-8B and Mistral-7B. The lower model is initialized with the weights of the first $M$ layers from the same LLM, where we set $M = 4$ in language modeling and $M = 16$ in SFT. We train SHAREDLLM on an $8\times$ A800 GPU machine. The batch size is set to 1 per GPU with gradient accumulation of 16 steps (global batch size is 128) for language modeling and 1 step (global batch size is 8) for SFT. The cross-attention layers remain fully tunable, while we opt to train the upper model's top $N - M$ self-attention layers in language modeling as post-injection aggregation for faster convergence.

**Baseline Methods.** For post-pretraining, we compare with other baselines in the same category which have extrapolation abilities, such as Positional Interpolation (Chen et al., 2023), YaRN (Peng et al., 2023) and CEPE (Yen et al., 2024). For SFT, we additionally compare with training-based methods, like StreamingLLM (Xiao et al., 2024b), LongAlpaca (Chen et al., 2024), and Activation Beacon (Zhang et al., 2025), as well as the advanced inference time method, SnapKV (Li et al., 2024) and OmniKV (Hao et al., 2025).

### 3.2 MAIN RESULTS

We first report the results on language modeling at various input lengths, which compares the extrapolation (length generalization) capability among methods. All perplexity values reported in Tables 1 and 2 are averaged over 1000 examples, except for the 128K length on which we test only 10 examples due to the data scarcity (Yen et al., 2024; Zhang et al., 2025). The results unveil our model's strong extrapolation capability—it successfully avoids perplexity explosion even when tested at the 128K-token length, though only having seen up to 8K-token sequences during training. Notably, SHAREDLLM outperforms CEPE in nearly all cases except the run at 128K tokens on ProofPile, showcasing the effectiveness of the introduced self-injection mechanism. Moreover, the improvement over Activation-Beacon is more pronounced than over CEPE, as CEPE experiences an additional pretraining stage and a warmup stage to align the hidden space between its encoder and decoder. In contrast, SHAREDLLM can directly be finetuned from publicly available *off-the-shelf* checkpoints, which significantly reduces training costs.

**Long-context Understanding Benchmarks.** We continue to test the supervised fine-tuned version of SHAREDLLM on tasks from LongBench (Bai et al., 2023) and InfiniBench (Zhang et al., 2024b). The two benchmarks comprise a variety of long-context tasks and cover various input lengths, which help to quantify both task and length generalizability in a unified manner.

For LongBench, we report the average scores on all 14 English tasks from 5 categories, including **single-document QA (SD-QA), multi-document QA (MD-QA), summarization (Summ.), few-shot learning/reasoning (FS) and code-completion (Code)**. For InfBench, we report the results on three representative tasks: Mathematical Find (Math.F), English Multi-Choice (EN.MC) and Retrieval of Numbers (Ret.N). SHAREDLLM outperforms or matches other advanced instruction-tuned long-context baselines across all five categories. In Table 3, SHAREDLLM surpasses advanced baselines on both benchmarks, showing superior capabilities in tackling extremely long inputs. We note that middle-truncation, a common practice in previous works, could reduce the difficulty of some tasks and improve the performance (Zhang et al., 2025), especially on decoder-only models,

Table 1: Language modeling results (perplexity) of the **continual pretraining setting** on downsampled RedPajama. Best results on *context-extended* models are marked in bold. Perplexity higher than $10^2$ are denoted by dash ("-"). LLaMA-3.1 has the declared 128K context-length since release, and we list the direct inference results separately for reference only.

| Base Model | Arxiv | | | | PG19 | | | | ProofPile | | | |
|---|---|---|---|---|---|---|---|---|---|---|---|---|
| | 4K | 8K | 32K | 128K | 4K | 8K | 32K | 128K | 4K | 8K | 32K | 128K |
| LLaMA-2-32K (Together, 2023) | 3.58 | 3.34 | 2.96 | OOM | 6.93 | 6.81 | 7.04 | OOM | 2.87 | 2.58 | 2.47 | OOM |
| PI (Chen et al., 2023) | 3.49 | 3.21 | 2.77 | OOM | 6.97 | 6.77 | 6.89 | OOM | 2.77 | 2.64 | 2.51 | OOM |
| YaRN (Peng et al., 2023) | 3.35 | 3.09 | 2.58 | OOM | 6.85 | 6.62 | 6.91 | OOM | 2.82 | 2.56 | 2.47 | OOM |
| CEPE (Yen et al., 2024) | 3.03 | 3.02 | 2.51 | 2.97 | 6.69 | 6.40 | 6.80 | 6.10 | 2.38 | 2.43 | 2.45 | **2.39** |
| SHAREDLLM | **2.99** | **2.97** | **2.46** | **2.91** | **6.55** | **6.28** | **6.65** | **5.96** | **2.33** | **2.34** | **2.38** | 2.40 |
| LLaMA-3.1 | 3.17 | 3.26 | 2.63 | 3.12 | 6.77 | 6.52 | 6.84 | 6.03 | 2.58 | 2.54 | 2.52 | 2.48 |

Table 2: Langauge modeling results of the **supervised fine-tuning** setting. "OOM" denotes the out-of-memory exception is raised during inference. Excessively large perplexities ($> 10^2$) are hidden with a dash ("-").

| Base Model | Method | PG19 | | | | ProofPile | | | | CodeParrot | | | |
|---|---|---|---|---|---|---|---|---|---|---|---|---|---|
| | | 4K | 16K | 32K | 100K | 4K | 16K | 32K | 100K | 4K | 16K | 32K | 100K |
| LLaMA-2 | StreamingLLM | 9.21 | 9.25 | 9.24 | 9.32 | 3.47 | 3.51 | 3.50 | 3.55 | 2.55 | 2.60 | 2.54 | 2.56 |
| | LongAlpaca-16K | 9.96 | 9.83 | - | OOM | 3.82 | 3.37 | - | OOM | 2.81 | 2.54 | - | OOM |
| | Activation Beacon | 9.21 | 8.34 | 8.27 | 8.50 | 3.47 | 3.34 | 3.32 | 3.31 | 2.55 | 2.43 | 2.41 | 2.62 |
| | SHAREDLLM | **8.68** | **8.01** | **7.96** | **8.24** | **3.36** | **3.24** | **3.21** | **3.19** | **2.33** | **2.25** | **2.23** | **2.36** |
| Mistral-7B | StreamingLLM | 9.58 | 9.63 | 9.52 | 9.55 | 4.08 | 4.19 | 4.16 | 4.23 | 2.99 | 3.05 | 3.13 | 3.02 |
| | LongAlpaca-16K | 10.21 | 10.39 | - | OOM | 3.26 | 3.34 | - | OOM | 3.05 | 3.21 | - | OOM |
| | Activation Beacon | 9.35 | 9.41 | 9.39 | 9.48 | 3.82 | 3.64 | 3.69 | 3.72 | 2.96 | 2.85 | 2.74 | 2.92 |
| | SHAREDLLM | **8.97** | **9.02** | **8.98** | **9.05** | **3.58** | **3.38** | **3.49** | 3.74 | **2.71** | **2.68** | **2.58** | **2.76** |

as the relevant information for many tasks is located at the head or rear of the entire context rather than the middle part.

## 3.3 TIME AND MEMORY EFFICIENCY

SHAREDLLM shows high computational efficiency in terms of both speed and GPU memory utilization. As Figure 3 visualizes, we compare the average inference time (ms) and memory consumption (GB) produced by SHAREDLLM against other advanced baseline models from the architecture types of streaming (Zhang et al., 2025), encoder-decoder (Yen et al., 2024) and vanilla with positional encoding (Peng et al., 2023) that have shown competitive

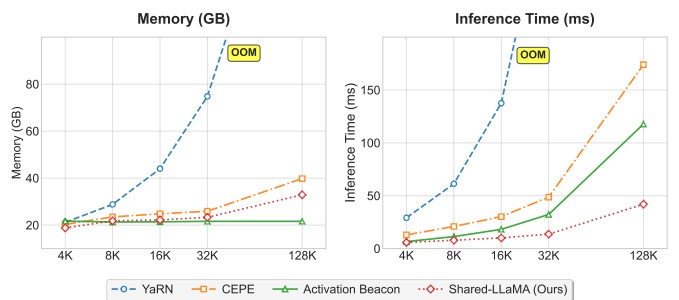

Figure 3: Comparison of memory usage (left) and total inference time on 100 examples (right) between SHAREDLLM and other **training time** baseline methods. The data is collected by running a tiny experiment on 100 examples in corresponding lengths. "OOM" means out-of-memory exception triggered during test time.

performance in prior evaluations. YaRN (Peng et al., 2023), which exploits the same fully attention as vanilla auto-regressive LLaMA, has $O(L^2)$ time and space complexity. The squared complexity makes it the only model that triggers the out-of-memory (OOM) exception at 128K length. Activation Beacon (Zhang et al., 2025), which adopts the streaming processing paradigm, maintains a minimum constant memory $O(l)$ under different input lengths $L$, where $l$ is the sliding window length, a predefined constant hyperparameter. However, Activation Beacon is incompatible with

Table 3: Evaluation results of different SFT methods on two benchmarks from LongBench and Infini-Bench. Note that for some baselines we follow their default settings to truncate the input below their window length, which may cast positive effects on their performance.

| Base Model | Base Model | LongBench | | | | | InfBench | | |
|---|---|---|---|---|---|---|---|---|---|
| | | SDQA | MDQA | Summ. | FS | Code | Math.F | En.MC | Ret.N |
| LLaMA-2 | Base | 24.90 | 22.60 | 24.70 | 60.00 | 48.10 | 2.85 | 22.79 | 1.85 |
| | StreamingLLM | 21.47 | 22.22 | 22.20 | 50.05 | 48.00 | 6.00 | 32.31 | 5.23 |
| | LongAlpaca-16K | 28.70 | 28.10 | **27.80** | **63.70** | 56.00 | 6.23 | 25.74 | 4.87 |
| | SnapKV | 24.05 | 22.98 | 17.25 | 16.11 | 58.87 | 9.95 | 28.83 | 2.31 |
| | OmniKV | 23.86 | 22.77 | 21.09 | 35.74 | 49.37 | 8.81 | 26.25 | 3.66 |
| | Activation Beacon | 28.27 | 28.44 | 25.15 | 61.00 | 57.75 | 12.14 | 32.05 | 80.58 |
| | SHAREDLLM | **28.83** | **30.93** | 25.76 | 63.50 | **59.93** | **13.82** | **33.65** | **82.79** |
| Mistral-7B | Base | 23.10 | 16.20 | 23.17 | 48.20 | 46.10 | 3.57 | 20.65 | 5.41 |
| | StreamingLLM | 26.19 | 16.65 | 23.48 | 48.23 | 45.98 | 7.26 | 18.84 | 9.75 |
| | LongAlpaca-16K | 27.05 | 17.33 | 26.18 | 51.97 | 52.28 | 5.41 | 21.19 | 12.48 |
| | SnapKV | 22.87 | 16.43 | 16.47 | 19.74 | **54.09** | 4.73 | 16.18 | 15.71 |
| | OmniKV | 22.95 | 16.87 | 21.36 | 42.85 | 41.90 | 3.81 | 19.77 | 14.98 |
| | Activation Beacon | 29.89 | 18.04 | 25.92 | 52.36 | 51.81 | 14.72 | 28.71 | 62.37 |
| | SHAREDLLM | **30.75** | **19.81** | **27.43** | **54.92** | 53.74 | **16.12** | **29.80** | **65.73** |
| LLaMA-3 | Base | 5.12 | 7.95 | 26.13 | 68.75 | 56.04 | 9.93 | 24.17 | 49.85 |
| | StreamingLLM | 6.73 | 8.56 | 26.85 | 68.32 | 54.83 | 11.27 | 35.81 | 52.85 |
| | LongAlpaca-16K | 21.41 | 12.45 | 27.74 | 70.72 | 60.05 | 12.03 | 25.28 | 16.13 |
| | SnapKV | 3.31 | 6.52 | 19.96 | 21.05 | **66.71** | 7.82 | 17.73 | 43.51 |
| | OmniKV | 4.54 | 8.21 | 20.77 | 32.19 | 57.92 | 8.29 | 21.16 | 41.10 |
| | Activation Beacon | 22.08 | 13.75 | **29.06** | 70.67 | 61.14 | 15.56 | **37.17** | 95.18 |
| | SHAREDLLM | **22.62** | **14.32** | 28.94 | **71.45** | 63.57 | **17.26** | 36.99 | **97.31** |

FlashAttention (Dao, 2023) also due to its specialized attention paradigm, which causes a sharp increment in inference time as input size grows. CEPE can process past context chunks in parallel, but these chunks must be passed through all its encoder layers (24-layer RoBERTa in CEPE) and layer-wise linear projections to obtain the final hidden states for cross-attention, leading to even slower inference speed than non-parallel Activation Beacon. In contrast, SHAREDLLM avoids such redundancy through shallow-layer compression and injection, which exhibits significant speed-up and limited memory consumption.

## 3.4 ABLATION STUDY

**Validation of Design Choices.** We conduct more experiments on the following ablative settings to validate the rationale behind the design choices: 1) the choice of context information injection layers; 2) other configurations, including the effect from the contextual information collection policy $\pi$ (only for instruction-following tasks), the noise in node splitting, and the addition of chunk-level positional indices during cross-attention. Regarding the layers selected to transmit KV cache for cross-attention, our implementation, which adapts the *continuous bottom* strategy and injects the context information in the bottom $M$ layers, obtains the strongest performance over the other two choices, not to mention its outstanding efficiency from the shortest forwarding and back-propagating path. For other settings, as shown in the bottom rows, performance significantly drops after removing any of the three items. Among the three items, the query-aware information gathering mechanism plays the most crucial role, as removing it causes the largest performance drop on query-driven tasks. In addition, the decoder's awareness of the sequential order of chunks is essential since the key-value pairs produced by the encoder are fed in a shuffled manner and must be accurately ordered via positional indices. Finally, introducing noise serves as an effective regularizer during training and also contributes to improved overall performance.

**Architecture Hyperparameters.** We further examine SHAREDLLM's sensitivity to some key hyperparameters, such as **tree height** and **token compression ratio**. The performance fluctuation on the same two tasks across these configurable hyperparameters are depicted in Figure 4. The figure reflects the sensitivity to hyperparameters, indicated by the inconsistent trend when tree hight is less than 3 and compression ratio is smaller than 8.

Table 4: Ablative Studies on different configurations of structural information injection. The best values in each category and settings consistent with our defaults are highlighted in **bold**.

Figure 4: Results on arxiv-32K (perplexity) and MD-QA (average F1) when configuring with different tree heights (left) and compression ratios (right) to SHAREDLLM. The values on the horizontal axis represent these individual variables. The value from the default configuration are highlighted in **bold**.

| Configuration | arxiv | MD-QA |
|---|---|---|
| **Default** | **2.46** | **30.93** |
| Continuous Top | 2.61 | 28.66 |
| Interleaving | 2.57 | 29.15 |
| w/o query-aware | - | 29.27 |
| w/o noise | 2.51 | 30.08 |
| w/o chunk pid | 2.49 | 29.81 |

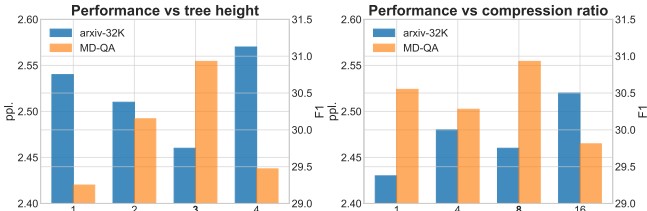

The left bar chart reveals the importance of the proper tree height. If the height is excessively small, then the tree is *undersplit* and the chunk size is so large that only coarse-grained information is preserved and received by the upper model, while task-related fine-grained information is lost. Conversely, if the tree is too high, then the tree is *oversplit* and the leaves carry minor details, which are less useful for tasks demanding a global view of the context, the downstream performance also degrades significantly. A similar trend can be captured with the global compression ratio $\beta$. Although the perplexity declines when all KVs ($\beta = 1$) are retained for cross-attention as more semantic information can be utilized, the query-aware information gathering ability deteriorates and thus the MDQA score becomes lower.

Besides the effect on task performance, we also conduct more experiments to explore how these configurations impact speed and memory in Appendix C.

## 4 RELATED WORK

**Building Long-context Language Models.** There are two prevalent routines to empower LLMs' capability to process extremely long text: directly pretraining on long-context corpus (Touvron et al., 2023a; Dubey et al., 2024; Jiang et al., 2023; GLM et al., 2024; Yang et al., 2025) or adapting short context-window LLMs to longer context lengths via combined various techniques (Tworkowski et al., 2024). The former approach consumes enormous amounts of data and computational resources, while the latter makes room for researchers to explore more flexible optimization strategies (Fu et al., 2024). Adaptation methods intend to *mimic* short input scenarios when the input text is actually long. Typical implementations include positional encoding (PE) rescaling (Press et al., 2021; Chen et al., 2023; Peng et al., 2023) and positional index rearranging (Xiao et al., 2024b; Ding et al., 2023; An et al., 2024; He et al., 2024). Both adjust the attention weight distribution to resemble the short-input scenarios. Another line of work compresses past tokens sequentially into dense representations (Chevalier et al., 2023; Zhang et al., 2025; Gao et al., 2025), serving as next-step input or storing them in an *external* retrievable memory (Wu et al., 2022; Xiao et al., 2024a). Yen et al. (2024) utilizes small model (Liu, 2019) for context compression to enable higher parallelism and minimize latency. However, this heterogeneous architecture necessitates extra pretraining and warmup stages to stabilize the fine-tuning process. Packer et al. (2023) introduces a system-level approach that introduces a hierarchical architecture along with a predefined set of I/O operations, enabling LLMs to offload, store, and retrieve long-range contextual information while maintaining a bounded active context within the model's window. Nevertheless, the effectiveness of this mechanism is fundamentally constrained by the capability of the underlying backbone LLM. In contrast to these works, our method directly tunes *off-the-shelf* models to compress context into structural representations for query-aware retrieval. Powered by efficient architecture design and a fast-forwarding mechanism, the whole procedure can be fully paralleled online without excessive memory usage.

**Efficient Techniques for Long-context Modeling.** In vanilla self-attention, the space and time complexity grows quadratically ($O(L^2)$) with the input sequence length $L$, which usually causes

out-of-memory (*OOM*) issues on GPU clusters when inputs are extremely long. A straightforward solution is to add parameter efficient fine-tuning (PEFT) modules (Chen et al., 2024; Zhang et al., 2025; 2024a) to shrink the size of gradient tensors during back-propagation. Many works strive to reduce the memory footprint of attention computation to enhance computational efficiency. Longformer (Beltagy et al., 2020) introduces a hybrid attention pattern to capture local and global semantic features concurrently. (Katharopoulos et al., 2020) designs linearized attention that merely demands $O(L)$ space to accomplish attention computation. FlashAttention (Dao et al., 2022; Dao, 2023) and PagedAttention (Kwon et al., 2023) maximize the memory efficiency from the system's perspective. More recently, (Xiao et al., 2024b) discovers the "attention sink" phenomenon and constructs pseudo sink to address the issue under window-attention. Similar attention patterns have been identified in (Han et al., 2024; Ge et al., 2024; Zhang et al., 2025) and leveraged as a principle when sparsifying attention maps during long-context modeling. Our work basically follows the efficient design principle in three aspects: 1) lightweight architecture through lower-layer self-injection; 2) compact representations via structural information extraction and compression; 3) efficient construction and retrieval algorithm based on context tree data structure.

## 5 CONCLUSION

In this work, we present SHAREDLLM, a novel and highly efficient framework that leverages a self-injection mechanism to adapt off-the-shelf short-context LLMs for long-context modeling. To overcome the computational bottlenecks of standard transformers, we integrate multi-grained context compression and query-aware information retrieval into a dedicated dynamic context tree structure. This allows the model to selectively isolate fine-grained details for relevant chunks while maintaining coarse-grained summaries for others. Extensive experiments demonstrate that SHAREDLLM not only successfully extrapolates to sequences exceeding 128K tokens but also excels across various language modeling and downstream instruction-following tasks. Crucially, it achieves this while maintaining superior memory and inference time efficiency compared to existing streaming and encoder-decoder architectures. Furthermore, because SHAREDLLM utilizes shared key-value mechanisms and is initialized from the same foundational layers, it can be directly fine-tuned from existing pre-trained checkpoints. This eliminates the need for costly post-pretraining or complex feature alignment stages, offering a highly accessible learning paradigm. Looking ahead, we hope this paradigm can be broadly generalized to other short-context LLMs. Future work will explore extending this scalable context-window extension approach to more complex architectural paradigms, such as multimodal large language models, ultimately paving the way for efficient, infinite-context processing across diverse data modalities.

## ETHICAL STATEMENT

All datasets in this paper are publicly available and have been widely tested in previous works. We do not leverage any synthetic data during training or evaluation. Components in SHAREDLLM are initialized from the checkpoint of released open-sourced LLMs and its security has been sufficiently validated when input queries are safe. We also scrutinized many sampled outputs and found no harmful information was generated.

## REPRODUCIBILITY STATEMENT

We provide many materials for reproduction in the appendix, including training and testing configurations, pseudo code snippets, and an anonymous code repository, etc. More evidence, such as the full code and model checkpoints, will be released in a later time.

## ACKNOWLEDGMENT

This research was supported in part by NUS Start-up Grant A-0010106-00-00. This work was also supported by the National Natural Science Foundation of China under Grant No. 62320106007.

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

## USAGE OF LLMs

We only use LLMs for writing suggestions, and revising purposes, including basic spelling, grammar, polishing, and LaTeX code formatting. The major workloads in this paper, such as ideation, coding, experiments, and paper writing, are fully completed by ourselves, while leaving LLMs as an auxiliary assistant instead of a major contributor to this paper.

## A   MORE IMPLEMENTATION DETAILS

### A.1   TRAINING CONFIGURATIONS

In accord with the settings in previous works (Chen et al., 2024; Yen et al., 2024; Zhang et al., 2025), for continual pretrainng, we initialize both lower and upper model with the base version (pretrained, non-finetuned) of LLMs. In SFT, we use their corresponding instruction-tuned version as the start point for training.

Table 5: Configurations for training on both tasks.

| Item | Continual Pretraining | Supervised Fine-tuning |
|---|---|---|
| training epoch | 1 | 2 |
| warmup ratio | 0.01 | 0.001 |
| $\sigma$ | 1/5 | 1/10 |
| chunk size | 1024 | 512 |
| $\alpha$ | \multicolumn{2}{c}{$1/16, 1/8, 1/4$} |
| AdamW $(\beta_1, \beta_2)$ | \multicolumn{2}{c}{0.9, 0.999} |

Zero Redundancy Optimizer (ZeRO) stage 3 from DeepSpeed without offload is enabled in both training and inference to allocate the memory usage evenly among GPUs. The cross-attention layers remain fully tunable, while we opt to train upper model's top $N - M$ self-attention layers in language modeling as post-injection aggregation for faster convergence. No parameter efficient fine-tuning (PEFT) techniques, such as LoRA, are applied during the training time, as PEFT seriously slows down model's convergence (Chen et al., 2024), which consequently costs longer tuning time than partial parameter fine-tuning. We adopt AdamW optimizer with the starting learning rate $1e^{-5}$ and cosine scheduler during training.

We list more training configurations that are not specified in the main text in Table 5. The sequential values of $\alpha$ are level-wise compression ratio, from level 1 to level 3.

### A.2   DATASET STATISTICS

We use different compositions of training dataset in continual pretraining and supervised fine-tuning below.

**Downsampled Redpajama.**   We follow (Yen et al., 2024) and (Touvron et al., 2023b) to prepare our training set. The proportions of data regarding seven domains in the resulted training set are listed in Table 6. All documents are truncated by 8,192 tokens to fit in the pretraining mode.

Table 6: Dataset composition in our downsampled Redpajama (10B) tokens.

| Domain | Proportion (%) |
|---|---|
| Arxiv (Clement et al., 2019) | 2.5 |
| Books (w/o S3) (Rae et al., 2020) | 4.5 |
| C4 (Roberts et al., 2019) | 15.0 |
| CommonCrawl (Luccioni & Viviano, 2021) | 67.0 |
| Github | 4.5 |
| StackExchange | 2.0 |
| Wikipedia (Foundation) | 4.5 |

During pretraining, 4K tokens are fed to the lower model and upper model respectively. The language modeling loss is calculated on the upper model's token prediction.

**Mixed Dataset in SFT.** This dataset is directly picked from (Zhang et al., 2025), which is a mixture of RedPajama and LongAlpaca (Chen et al., 2024). LongAlpaca is composed of Stanford Alpaca instruction-following dataset (Taori et al., 2023) and author-curated long-context tasks such as summarization and long-document question answering. We follow (Zhang et al., 2025) to filter samples and only preserve those whose lengths range from 1K to 8K. The distribution of samples in terms of length is specified in Table 7.

Table 7: Proportion of samples within each length interval.

| Length | <2K | 2~4K | 4~6K | 6~8K |
|---|---|---|---|---|
| Proportion | 47% | 29% | 8% | 16% |

Since we found there was an absence of training data in fine-grained retrieval tasks, we additionally sample a small set (200 samples) of data from Llama-3-8B-262K training corpus and add them to the SFT data collections. This tiny proportion of data plays decisive roles in ensuring SHAREDLLM's non-decreasing accuracy as the input context length grows.

### A.3 ONLINE SPLIT-AND-SEARCH ALGORITHM

We provide the pseudo code for the online split-and-search algorithm introduced in Section 2.2, from the splitting of the root node till collecting all key-value states for all preserved nodes and all $M$ layers. The full implementation is not intricate, which can be readily accomplished with around 25 lines of code.

For the full set of the core code, please refer to `https://anonymous.4open.science/r/sharedllm_anony-04B1` for details. The code snippet in the entire model.py file can also be found in this anonymous repository.

---

**Algorithm 1** Pseudo code of dynamic Construction-and-Search.

```
# N: number of trees; L: chunk size

# depth: tree depth; chunk_ids: the entire input ids for chunk in shape (N, L)
# gamma: a hyper-parameter to adjust the variance of the gaussian sampling

selected_input_ids = chunk_ids
selected_length = chunk_ids.shape[-1]
all_kvs = []

for i in range(depth):
    # sample lengths of left and right child
    if i < depth - 1:
        half_length = last_length // 2
        sigma = half_length / gamma
        delta = random.randn(1) * sigma
        l_left, l_right = half_length - int(delta), half_length + int(delta)

        # split the node into two children
        left_input_ids, right_input_ids = input_ids[:l_left], input_ids[-l_right:]
        # query_aware is a flag indicating if the selected nodes are determined on query
        if query_aware:
            # short forward (1-layer) to get representation vectors for the query and two nodes
            h_q = upper_model(query, 1)
            h_left, h_right = lower_model(left_input_ids, 1), lower_model(right_input_ids, 1)
            selected = argmax(sim(h_q, h_left), sim(h_q, h_right))
        else:
            selected = 1 # deterministic example, can change to 0 or random selection

        selected_input_ids = [left_input_ids, right_input_ids][selected]
        selected_length = [l_left, l_right][selected]

        preserved_input_ids = [left_input_ids, right_input_ids][1 - selected]
    else:
        preserved_input_ids = cat(last_input_ids.chunk(2, -1), 0)

    cur_level_kvs = lower_model(preserved_input_ids).past_key_values
    cur_level_kvs = downsample(cur_level_kvs)
    all_kvs.append(cur_level_kvs)
```

---

`cat`: concatenation; `chunk`: split into the specified number of chunks

---

## A.4   Consequence from the absence of Book-S3

Book-S3 is a large dataset of copyrighted published books composed by professional writers in various domains. Due to the copyright infringement allegations, all online entries to access this corpus have been removed. Prior studies (Yen et al., 2024) have shown that the absence of Book-S3 subsets in RedPajama corpus casts a negative impact on language modeling results. Here we simply show the comparison in terms of perplexity when SHAREDLLM is trained with and without Book-S3. As Table 8 shows, the baselines without Book-S3 as part of their continual pretraining corpus show inferior results, which is consistent with the observation in Yen et al. (2024). We hypothesize that the root cause is that Book-S3 contains many well-structured and logically sound articles written by expert-level writers, which show higher quality and lower noise than data from other domains. Therefore, it plays a great role in improving language modeling.

Table 8: Perplexity increment as a negative effect from the lack of books3. † represents the values in corresponding rows are reproduced from open-sourced code.

| Model | Arxiv | | | | PG19 | | | | ProofPile | | | |
|---|---|---|---|---|---|---|---|---|---|---|---|---|
| | 4K | 8K | 32K | 128K | 4K | 8K | 32K | 128K | 4K | 8K | 32K | 128K |
| LLaMA-2-7B (4K) | 2.60 | - | - | OOM | 6.49 | - | - | OOM | 2.28 | - | - | OOM |
| *Books3 involved in training* | | | | | | | | | | | | |
| YaRN-2-128K | 3.13 | 2.96 | 2.34 | OOM | 6.15 | 6.02 | 6.32 | OOM | 2.70 | 2.47 | 2.41 | OOM |
| CEPE | 2.86 | 2.84 | 2.34 | 2.91 | 6.60 | 6.24 | 6.66 | 5.99 | 2.22 | 2.33 | 2.26 | 2.23 |
| *Books3 not involved in training* | | | | | | | | | | | | |
| YaRN-2-128K | 3.46 | 3.30 | 2.57 | OOM | 6.83 | 6.59 | 7.14 | OOM | 2.85 | 2.68 | 2.63 | OOM |
| CEPE† | 3.03 | 3.02 | 2.51 | 2.97 | 6.69 | 6.40 | 6.80 | 6.10 | 2.38 | 2.43 | 2.45 | **2.39** |
| SHAREDLLM | **2.99** | **2.97** | **2.46** | **2.91** | **6.59** | **6.31** | **6.72** | **6.00** | **2.36** | **2.37** | **2.41** | 2.46 |

## A.5   Details of Test Benchmarks

For all inference results, we report the average values of five runs under different random seeds.

**RedPajama**   To test the long-context modeling capability, we use a tiny proportion of corpus which has never been seen by the model during the continual pretraining period as the test set. The sampled passages are ensured to match the corresponding test lengths. We hold a constant 4K input for the upper model while the left long context is passed to the lower model, akin to what we did during pretraining.

**Long Bench**   (Bai et al., 2023) is the first bilingual (English and Chinese), multi-task benchmark for long context understanding. It comprises 21 datasets (16 English and 5 Chinese) across 6 subcategories, which aims for a more rigorous evaluation of long context understanding. These categories encompass *single document QA, multi-document QA, summarization, few-shot learning, synthetic tasks, and code completion*. The average length of documents is 6,711 words in English and 13,386 characters in Chinese.

**Infinity-Bench**   (Zhang et al., 2024b) extends context lengths in previous long-context benchmarks from 10K to more than 100K tokens. The benchmark is composed of synthetic and realistic tasks that span diverse domains and bilingual (Chinese and English), such as retrieval (Ret.), summarization (sum), question answering (QA), code and math.

# B   More Experimental Results

## B.1   Resutls on Passkey Retrieval

We further assess the retrieval capability of SharedLLM on the passkey retrieval, or needle-in-haystack (NIAH) task. Following the settings in (Yen et al., 2024), we train a new version of

SharedLLM that can perform accurate passkey retrieval from the haystacks of the surrounded non-sense. We follow the examples in (Chen et al., 2024) to set up the single key-value pair test cases. The results averaged on 10 randomly generated NIAH test samples are shown in Figure 5. It can be observed that SHAREDLLM enjoys the minimal accuracy decay as length extends compared to other baselines, although it has only seen context within 8K length.

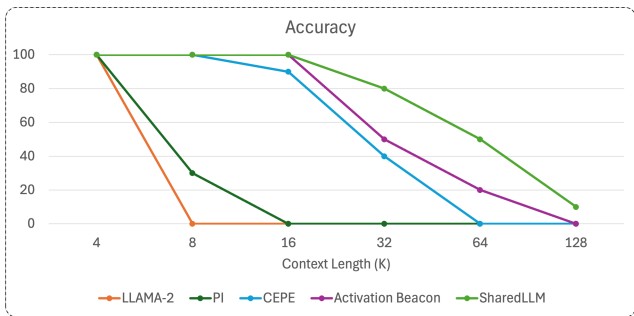

Figure 5: Accuracy comparison on passkey retrieval (single key-value pair) task.

## B.2 COMPARISON BETWEEN DIFFERENT ATTENTION MAPS

The introduced self-injection algorithm can also be regarded as an attention remapping process, where we use "continuous-right" and "query-aware" node selection strategy for language modeling and long-context understanding respectively. Meanwhile, many concurrent works (Han et al., 2024; Xiao et al., 2024b; Ge et al., 2024) observed the special $\Lambda$-shape attention map and took advantage of this for acceleration. In fact, the policy selection is not only intuitive but also with the fundamental support from pilot experiments. We report the results of all these choices below:

Table 9: Pilot studies of branch-selection policies.

| Setting | Arxiv | MD-QA |
|---|---|---|
| Default | **2.46** (**±0.01**) | **30.93** (±0.16) |
| Random | 2.61 (±0.13) | 28.85 (±0.45) |
| Fixed Left | 2.49 (±0.02) | 29.62 (±0.15) |
| Query-aware | 2.48 (±0.02) | 29.27 (±**0.12**) |
| $\Lambda$-shape | 2.52 (±0.04) | 29.48 (±0.18) |

The results manifest that the selected policy can produce the optimal performance on the downstream tasks.

## C TIME AND MEMORY EFFICIENCY

### C.1 QUANLITATIVE ANALYSIS

Apart from strong performance on downstream tasks, SHAREDLLM demonstrates high computational efficiency in terms of both inference speed and GPU memory utilization. We compare these metrics produced by SHAREDLLM against other representative models of streaming (Zhang et al., 2025), encoder-decoder (Yen et al., 2024) and vanilla (Peng et al., 2023) architectures that have shown competitive performance in prior evaluations. The results are visualized in Figure 3.

YaRN (Peng et al., 2023), which only modifies the encoding policy but still applies the vanilla multi-head attention as LLaMA, shows squared ($O(L^2)$) time and space complexity. Consequently, it triggers the out-of-memory exception at an early stage (128K tokens). Activation Beacon (Zhang et al., 2025), which adopts the streaming processing paradigm, maintains a minimum constant memory $O(l)$ under different input lengths $L$, where $l$ is the sliding window length. However, Activation Beacon is incompatible with FlashAttention (Dao, 2023) also due to its specialized attention

paradigm, which causes a sharp increment in inference time as input size grows. CEPE can process past context chunks in parallel, but these chunks must be passed through all its encoder layers (24-layer RoBERTa in CEPE) and layer-wise linear projections to obtain the final hidden states for cross-attention, leading to even slower inference speed than non-parallel Activation Beacon. In contrast, SHAREDLLM alleviates such redundancy through shallow-layer compression and injection, which exhibits significant speed-up and limited memory consumption.

We have explained the outstanding efficiency of our model by comparing the memory usage and inference speed with other competitors. In this section, we give a more comprehensive analysis towards the inherent factors that may impact model's efficiency, including compression ratio $\beta$, tree height $h$, the number of shared layers $M$ and the retrieval-based policy which requires an additional short forward pass.

Table 10: Inference time under various $M$ with constant $h = 3$ and $\beta = 8$. Our default setting is highlighted in **bold**.

| $M$ | 1 | 2 | **4** | 8 | 16 |
|---|---|---|---|---|---|
| Time (s) | 6.78 | 9.35 | **11.81** | 16.81 | 25.85 |
| Memory (GB) | 21.04 | 21.50 | **22.39** | 24.08 | 27.82 |

## C.2 EFFICIENCY RESULTS

We rerun our experiments to measure the forward time and memory cost from language modeling on 8K tokens, adjusting one variable at a time while keeping others at their default values. The results are shown in Table 10, 11 and 12. Among these factors, the number of injection layers, $M$, has the most significant impact on both speed and memory: both memory and latency grows as $M$ increases. As an opposite, compression ratio $\beta$ and tree height $h$ produces nuances effect on both metrics. For example, if we decreases $\beta$ from 64 to 1 (preserve all KVs), the inference time increases by 6.7% while memory increases by 3%. A similar trend is observed on experiments with tree height $h$. We speculate that the reason behind these outcomes are partly from the internal optimization in FlashAttention, which efficiently computes attention blockwisely. When the configuration meets its requirement for block size and hidden dimension (e.g., length is divisible by 256),

Table 11: Inference time under various $\beta$ with constant $h = 3$ and $M = 4$. Our default setting is highlighted in **bold**. For $\beta \in \{1, 2\}$, we are not able to set levelwise compression ratios and thus we set the compression ratio same as the $\beta$ for every level of the tree.

| $\beta$ | 64 | 32 | 16 | **8** | 4 | 2 | 1 |
|---|---|---|---|---|---|---|---|
| Time (s) | 11.68 | 11.73 | 11.78 | **11.81** | 11.87 | 12.04 | 12.47 |
| Memory (GB) | 22.20 | 22.20 | 22.20 | **22.39** | 22.40 | 22.35 | 22.97 |

Table 12: Inference time under various $h$ with constant $\beta = 8$ and $M = 4$. Our default setting is highlighted in **bold**.

| $h$ | 1 | 2 | **3** | 4 |
|---|---|---|---|---|
| Time (s) | 11.16 | 11.55 | **11.81** | 11.86 |
| Memory (GB) | 19.72 | 22.42 | **22.39** | 22.41 |

We further investigate the potential overhead caused by the extra short forward path query-aware splitting-and-search algorithm. As shown in Table 13, we observe that it incurs around 15% overhead in both time and space. We believe this type of overhead can be further eliminated with more careful optimization of the implementation details.

Table 13: Comparison of time and memory consumption when query-based retrieval is incorporated/not incorporated in SHAREDLLM. $h$, $M$ and $\beta$ are fixed at the default values.

| Setting | Time | Memory |
|---|---|---|
| w/o query-aware retrieval | 11.81 | 22.39 |
| w query-aware retrieval | 13.18 | 25.44 |

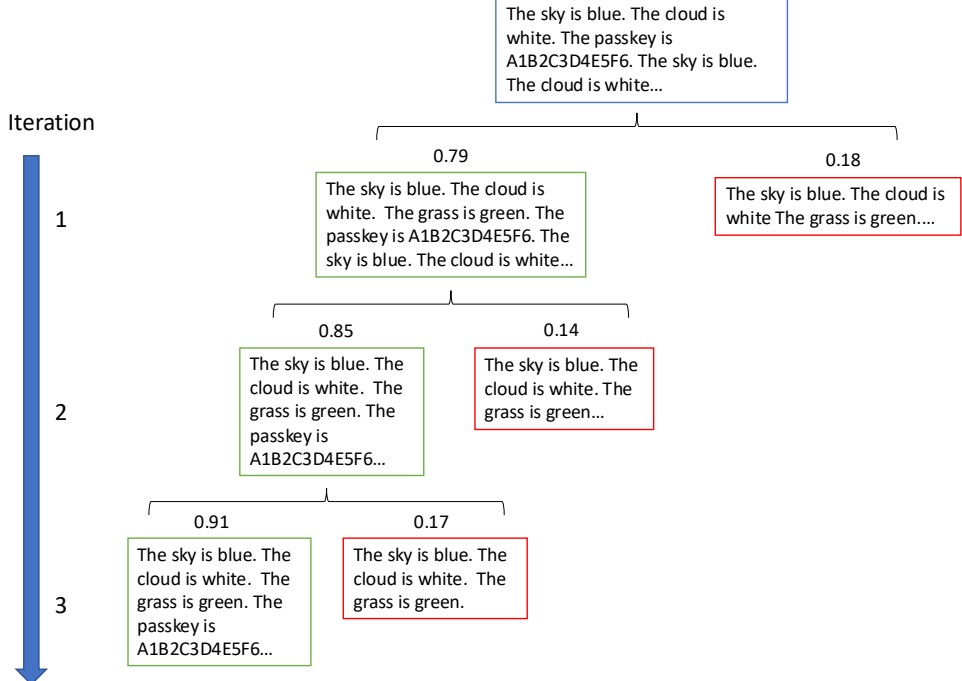

Figure 6: A living example of tree growth and split on the passkey retrieval task. The numbers above the text boxes are the correlation score between the text chunk and user query. Green/Red boxes indicate the chunk is selected/not selected.

## D VISUALIZATION OF TREE SPLITTING PROCESS

We provide a living example to demonstrate how the tree is constructed and how the key chunk is retrieved when performing the passkey retrieval task. In this example, we assume that the length of input text is 8,192 and the passkey is located between token id 15 and 20. The process is depicted in Figure 6. As the figure shows, SharedLLM first split the entire input of 4,096 tokens into two chunks of 2,048 tokens. Then, it computes the correlation scores between the query and subchunks, and finds the first chunk more correlated ($0.79 > 0.18$). Hence, it repeats the procedure by splitting that chunk into two subchunks of 1,024 tokens. The process iterates until the maximum tree depth has been reached (suppose $d_{max} = 3$), where the chunk size is 512. At each iteration, the chunks where the passkey resides are always selected due to their higher correlation scores.

