# OpenReview forum: "Stacked from One: Multi-Scale Self-Injection for Context Window Extension"
_ICLR.cc/2026/Conference — ICLR 2026 Poster_

### Official Review · Reviewer_CrBs · 2025-10-30

**Soundness:** 3
**Presentation:** 3
**Contribution:** 3
**Rating:** 6
**Confidence:** 3

**Summary:**

To address the limitation of finite context windows in Large Language Models (LLMs), this paper proposes the SHAREDLLM framework. Grounded in the concepts of multi-grained context compression and query-aware information acquisition, SHAREDLLM consists of two stacked models derived from the same base short-context LLM: a lower "compressor" and an upper "decoder". It enables "self-injection" through shared key-value (KV) states at the lower layers, thus avoiding redundant computations. A core innovation is the context tree—a binary tree structure that dynamically encodes long context into coarse-to-fine representations, expanding only task-relevant nodes. Trained on 8K-length sequences, the model can generalize to 128K+ tokens. It outperforms baselines such as CEPE and Activation Beacon on language modeling (perplexity) and long-context understanding tasks (LongBench/InfiniBench), with an inference speed 2× faster than streaming architectures and 3× faster than encoder-decoder architectures.

**Strengths:**

The dynamic expansion of the context tree adapts to task requirements, balancing information retention and efficiency, representing a relatively novel idea. It provides researchers with a lightweight alternative, lowering the threshold for long-context research; efficiency improvements expand the application of LLMs in long-text scenarios.

**Weaknesses:**

The model is only tested on up to 128K tokens, yet it claims to "generalize to arbitrary lengths"—supplementary experiments on 256K tokens or theoretical analysis are needed to support this claim.Experiments only provide quantitative results without qualitative examples (e.g., cases where the model correctly identifies key information in passkey retrieval tasks), making it difficult to intuitively demonstrate advantages.

**Questions:**

Can experiments on 256K tokens be supplemented? If not, theoretical justification for the "arbitrary length generalization" claim is required.

---

> ### Author Response · Authors · 2025-11-15
> **Response to Reviewer CrBs**
>
> Thanks for your detailed feedback. Below is our response to the weakness and questions part.
>
> ## Lack of Case Study
> We provide a diagram below for passkey retrieval here, and add the formal version to the new submission in appendix D. Please check
>
> Suppose the passkey is located in token 15-20, then the splitting-and-search process can be described as
> ```
>       t{1-8192)
>       //       \
>     t{1-4096}   t{4097-8192}
>       //    \
> t{1-2048} t{2049-4096}
>     //  \
> t{1-1024} t{1025-2048}
> ```
> where t{i-j} denotes the token sequence from index $i$ to $j$, and "//" represent the branch we select
>
> ## Lack of experiments on longer inputs
> - For instruction-following tasks, Infinity-Bench already includes test inputs as long as up to 1M tokens.
> - For language modeling tasks, we extend Table 1 by reporting the perplexity on 256K context as below. SharedLLM retains low perplexity, similar to results on short context lengths.
>
>     | Base Mode | Arxiv | PG19 | ProofPile |
>     | ---- | ---- | ---- | ---- |
>     | LlaMA-2-32K | OOM | OOM | OOM |
>     | PI | OOM | OOM | OOM |
>     | YaRN | OOM | OOM | OOM |
>     | CEPE | 2.76 |  6.13 | 2.59 |
>     | **SharedLLM** | **2.68** | **6.05** | **2.40** |
>     | Llama-3.1-8B | >100 | >100 | >100 |

---

> ### Author Response · Authors · 2025-11-28
> **Gentle reminder**
>
> Dear Reviewer,
>
> This is a gentle reminder to kindly view our response in OpenView. We would greatly appreciate it if you could check whether our reply has addressed your concerns and let us know if any further clarification or results are required.
>
> Your timely feedback will help us ensure that all issues are resolved before the discussion period concludes.
>
> Best regards,
> Authors

---

### Official Review · Reviewer_xDgb · 2025-10-31

**Soundness:** 2
**Presentation:** 3
**Contribution:** 2
**Rating:** 4
**Confidence:** 4

**Summary:**

This paper proposes a two-LLM architecture for long-context inference. One LLM compresses the context, and the other LLM operates based on compressed context, achieving 2x speedup without hurting accuracy.

**Strengths:**

* A more end-to-end approach that also includes LLM training stage.

**Weaknesses:**

* This is a crowded space, and needs to be more embedded into related work to justify the novelty of this work.

**Questions:**

* The proposed method --- using LLM to compress context --- reminds me of MemGPT and generative agent paper, where they also uses the LLM to compress the context. How do you compare with these two approaches? I understand that you train the model end-to-end, but algorithm-wise is there any reason to believe your approach is better?
* Given the accuracy number of your system is close to token dropping based approach, and SnapKV, though it is a strong baseline, is not state-of-the-art based on NVIDIA's KVPress measurement (https://huggingface.co/spaces/nvidia/kvpress-leaderboard), I would suspect other baselines may have higher accuracy than SnapKV and be better than your approach. Is there any reason to believe that is not the case?
* Regarding training --- why SFT instead of RL?

---

> ### Author Response · Authors · 2025-11-15
> **Response to Reviewer xDgb**
>
> Thanks for the detailed feedback. Below is our reply to the weakness and question part
>
> ## Q1.1 Compare MemGPT and SharedLLM
> We carefully checked the details of MemGPT paper.
> Both works aim to address the limited context window issue, but they differ significantly in design philosophy, optimization scope, and target use cases.
>
> - **Optimization Directions (system level v.s. model level)**: MemGPT constructs a retrieval system with a hierachical memory storage module on top of existing LLMs (GPT3.5/4). Apart from task executor, LLM also serves as the memory manager to read/write memories without modifying the backbone model's architecture or parameters---the internal implementation is based on API and function calls.
>     In contrast, SharedLLM introduces a new architecture and directly trains an open-sourced LLM, optimizing the model's parameters to natively support long-context modeling.
> - **Target Applications**: MemGPT is primarily oriented toward chat applications by continuously compress messages and recall memories in a streaming pattern. To perform document understanding tasks, a external retriever (text-embedding-ada-002) must be integrated into the system to help construct the vector database, which introduces additional latency or memory cost. In contrast, SharedLLM trains a unified model to solve a large scope of general long-context tasks, such as language modeling, code completion, in-context learning, summarization, multi-document QA. No extra components are required when tested on these tasks.
> - **Processing Patterns (streaming v.s. parallel)**: Due to the characteristics of streaming messages, MemGPT uses FIFO queues and thus processes input/output in a streaming manner. SharedLLM compresses chunk-level context with higher parallelism.
>
> ## Q1.2 Justification of Proposed Method
> We believe the capaility of base LLMs is more fundamental. SharedLLM operates on the model instead of building applications on the model, which directly improves LLM's capability on downstream tasks. In contrast, MemGPT builds its system with API service, which may suffer from error propagation if the base model generates incorrect or hallucinated output.
>
> Besides, during inference, our model keeps the fixed set of components and holds "memories" inside itself, which is more efficient than MemGPT.
>
> ## Q2: Potentially Stronger Baselines than SnapKV
> Works on the KVPress leaderboard (including SnapKV and OmniKV) are in a different family from SharedLLM---they are inference-time approach, which assume a sufficient long context window LLM (such as Llama-3.1, Qwen-V2.5) is already there. They do not train the model anymore and only aim at minimizing the time/memory cost while keeping performance at the same level. Consequently, their performance and context window size is **upper-bounded** by the base LLM (see table 2, SnapKV shows similar performance as the base model).
>
> SharedLLM, similar to other peer works (Activation Beacon, CEPE, StreamingLLM), does not only save the inference time, but also endeavours to extending small context window LLMs (e.g., LLaMA-2 with 4K context window) into larger ones. Training on 8K context, we expect to see promising performance on the context as longer as possible.
>
> Based on the analysis above, we agree that there could be better baselines in the page you provide. Here we report KVzip (1st place on the leaderboard, the latest baseline in NIPS 2025) on Llama-2 as below:
> | Method | SD-QA | MD-QA | Summ. | FS | Code | Math.F | En.MC | Ret.N |
> | ---- | ---- | ---- | ---- |  ---- | ---- | ---- | ---- | ---- |
> | Base | 24.90 | 22.60 | 24.70 | 60.00 | 48.10 | 2.85 | 22.79 | 1.85 |
> | SnapKV | 24.05 | 22.98 | 17.25 | 16.11 | 58.87 | 9.95 | 28.83 | 2.31 |
> | KVzip | 24.58 | 22.86 | 19.84 | 33.75 | 51.29 | 8.58 | 27.61 | 3.39 |
> | SharedLLM | 28.83 | 30.93 | 25.76 | 63.50 | 59.93 | 13.82 | 33.65 | 82.79 |
>
> The results match our expectation---KVzip has similar performance as the base model, which is slightly better than SnapKV but falls behind training-based approaches (such as Activation Beacon and SharedLLM), since no extra training is performed. This also validates our claim in Q1.2 that the capability of backbone LLM is of great importance.
>
> ## Q3: SFT Instead of RL
> The target tasks are language modeling and instruction-following ones, instead of reasoning, tool using, where RL shines. Besides, most of the baselines experience only SFT, so we follow the same setup for fair comparison.

---

> > ### Comment · Reviewer_xDgb · 2025-11-28
> > **Changing to weak accept, see comments for more details**
> >
> > Thank you for your response. In general I do agree with the argument that training the base model gives larger potential room for improvement, so raising to weak accept.
> >
> > The key concern that holds me back to higher score is that I don't see a concrete example that inference-time methods clearly cannot handle that, and training the base LLM can resolve that.

---

> > > ### Author Response · Authors · 2025-11-28
> > > **A Gentle Reminder**
> > >
> > > Dear Reviewer,
> > >
> > > Here's a gentle reminder that we have provided the concrete examples you would like to see. Please check. Thanks!
> > >
> > > Best,
> > > Authors

---

> ### Author Response · Authors · 2025-11-28
> **Concrete Examples (results)**
>
> Thanks for the feedback! The examples are easy to find---since inference-time methods **are NOT able to extend the context window of base LLMs (i.e., they cannot handle extremely long input)**, their performance degrades when the input length exceeds the context window of the base model.
>
> We provide two examples here, one for language modeling and one for NIAH, both of which use **LLaMA-2-7B as the base model (context window 4K)**. We compare the results of SharedLLM with SnapKV, the representative inference-time method.
>
> 1. Language Modeling on arxiv (perplexity, **lower** is better)
> | Method | 4K | 8K | 32K |
> | ---- |  ---- | ---- | ---- |
> | SnapKV (inference-time method) | 3.23 | >100 | >100 | >100 |
> | SharedLLM (training method) | **2.99** | **2.97** | **2.46** |
>
> 2. NIAH (accuracy, **higher** is better)
> | Method | 4K | 8K | 32K |
> | ---- | ---- | ---- | ---- |
> | SnapKV (inference-time method) | 91 | 0 | 0 | 0 |
> | SharedLLM (training method) | **100** | **98** | **95** | **92** |
>
> When the input lengths exceed 4K in both tasks, performance of the inference-time method (SnapKV) plumets, indicating that it cannot handle inputs that exceeds the base LLM's context window size, while training methods can address this with minimal degradations even when the input is much longer (8x context window).

---

### Official Review · Reviewer_4Mtb · 2025-11-01

**Soundness:** 3
**Presentation:** 4
**Contribution:** 3
**Rating:** 8
**Confidence:** 4

**Summary:**

This paper proposes SharedLLM, a novel two-stage LLM framework designed to extend the context window of short-context language models. Specifically, it consists of 1) a lower model (i.e., compressor) that segments long input sequences and compresses each into hierarchical “context trees” and 2) an upper model (i.e., decoder) that retrieves relevant information from them. The mechanism is called self-injection. Cross-attention from query to the shared KV between the same base model’s layers efficiently injects the compressed/selected input context information. SharedLLM demonstrates long-context generalization (up to 128K tokens trained only on 8K) capability and outperforms baselines such as CEPE and Activation Beacon. In addition, the paper reports accelerated inference and memory usage savings.

**Strengths:**

* The idea of using a single model for two complementary purposes in long-context processing is novel. Leveraging the same base model to ensure compatibility is both intuitive and efficient.
* The context tree construction, which identifies relevant segments within each chunk without requiring an additional similarity computation module, is original and practical.
* The method avoids complex attention mechanisms, enabling the reuse of existing optimization techniques such as FlashAttention.

**Weaknesses:**

* As shown in Figure 4, performance improvements with respect to tree height and compression ratio appear somewhat inconsistent, suggesting potential sensitivity to hyperparameters. Furthermore, although the method is generally robust within moderate ranges, its reliance on rule-based policy selection and heuristic coarse-to-fine downsampling may limit task generalization.
* The lack of inter-chunk dependency modeling is understandable for parallelization and optimization efficiency, but it may introduce limitations in long context integration.
* The Passkey task might be too well aligned with SharedLLM’s query-aware design, making it an easier benchmark for this method. Additional justification or analysis would strengthen the empirical evaluation.

**Questions:**

* The proposed mechanism seems to be designed for the prefill phase; it may have a limited impact during decoding.
* When positional indices are added in the cross-attention module, are they implemented as sinusoidal position embeddings directly added to the key–value states, rather than as RoPE-style rotations?
* During fine-tuning, are only the cross-attention components trained while other parameters are frozen, or is the model fully fine-tuned?
* (suggestion) The near-half-randomness in context tree construction is interesting, but its benefits are not entirely clear. Given that later tokens within a chunk often implicitly encode earlier information, maybe a deterministic segmentation strategy, such as splitting at points of large neighboring vector similarity differences, would be better.

**Details Of Ethics Concerns:**

No concerns.

---

> ### Author Response · Authors · 2025-11-15
> **Response to Reviewer 4Mtb**
>
> We sincerely appreciate your insightful feedback and suggestions. Below we provide our response to the points raised in the weakness and question part:
>
> ## Weakness
> ### W1 (hyper-parameter sensitivity and task generalization ability)
> - The figure reflects the sentitivity to hyperparameters, indicated by the inconsistent trend when tree hight is less than 3 and compression ratio is smaller than 8. However, this outcome also demonstrates the effectiveness of the designed tree structure, which works with a certain amount of tree height and compress ratio: when the hierarchy is too shallow or the compression is too weak, the model degenerates into the vanilla LLM (tree height = 1, compression ratio = 1), hence offering limited improvements.
>
>   In our experiments, we start from a high tree and large compression ratio (e.g., height$\ge$3, compress ratio$\ge$8) to consider the efficiency and the diverse information density across layers. The left two bar groups in each subplot are included primarily for completeness and visualization and have not been really tried during model tuning time.
>
> - This selection policy is only different among language modeling (for non-chat model) and instruction following (for chat model). The task and length generalizability is verified in a broad range of long-context understanding tasks as shown in Table 3, where SharedLLM is tuned on a set of limited types/lengths of tasks but perform pretty well on many unseen downstream tasks with much longer prompt input.
>
> ### W2 (lack of inter-chunk dependency)
> We agree that every design choices have its trade-off and know there could be some scenarios that this learning paradigm fails. Therefore, we tested SharedLLM on a broad range of tasks to demonstrate its task and length generalizability.
>
> ### W3 (cherry picking of evaluated tasks)
> Apart from passkey retrieval, other types of tasks from LongBench and InfBench were also evaluated, including Multi-hop QA, Summarization, etc, whose formulation is quite different from Passkey and does not align well with the architecture (Table 3). SharedLLM exhibits promising results on these tasks as well.
>
> ## Questions
> ### Q1
> Yes, the design works mainly in the prefiiling time instead of decoding time. But it also benefits the decoding step---the number of KVs is significantly reduced, leading to faster and memory-efficient attention computation per token generation.
>
> ### Q2
> We do not use the additive sinusodial position embedding.
> They are implemented as matrix multiplication between keys (from the lower model) and queries (from the upper model), similar to RoPE.
>
> ### Q3
> For the upper model, we finetune the cross-attention modules in the first M layers, and all modules for the top N-M layers (see Appendix A.1). Other modules are kept frozen. Since we find such a setting ensures both effieciency and stability during training.
>
> ### Q4
> Thanks for the insightful suggestion! It sounds like the splitting process of the decision tree (choose the best attribute that brings the largest entropy decrement), and could potentially provide a more expressive hierarchical structure. We truly considered similar design choices, but ultimately opted for a simpler implementation to maintain efficiency, as the search space would expand substantially when incorporating multiple splitting choices.
> We agree this is a promising direction, and we will further explore it to identify a representation that is both efficient and flexible.

---

> > ### Comment · Reviewer_4Mtb · 2025-11-25
> >
> > Thank you for the response.
> > Please consider including these details in the revised manuscript.
> > As the original score is already high, I'd maintain my evaluation.

---

> > > ### Author Response · Authors · 2025-11-25
> > > **Response to Reviewer 4Mtb**
> > >
> > > Thanks for the timely feedback and we are delighted to have provided the information you need.
> > >
> > > We have incorporated these details in the latest version at the corresponding locations.:
> > >
> > > - W1.1: We separate section 3.4 into two paragraphs. The sensitivity analysis is inserted into the second paragraph and highlighted blue.
> > > - W1.2 & W2 & W3: We emphasize the application scenarios for both policies in Section 2.2, and explain how the task/length generalizability is proved in each experiment in section 3.2.
> > > - Q1: Add description in section 2.1.
> > > - Q2: Add applying RoPE (relative positional embedding instead of sinusodial) to Q and K in section 2.3.
> > > - Q3: Add reference to Appendix in section 3.1.

---

### Official Review · Reviewer_AWmy · 2025-11-04

**Soundness:** 2
**Presentation:** 1
**Contribution:** 2
**Rating:** 2
**Confidence:** 3

**Summary:**

**Summary**

This work presents SHAREDLLM, a novel framework addressing the challenge of efficient long-context inference. Its key innovation is a two-stage process involving a lower model, which compresses context chunks via a Context Tree structure, and an upper model, which decodes  successive tokens from the compressed KV-cache. This architecture enables an extended context window while enhancing inference efficiency.

**Strengths:**

**Strengths**

（1）This study presents a novel method for addressing the challenges of long-context inference, with its architectural details thoroughly elaborated.

（2）The efficacy of the proposed framework is demonstrated through experiments, which confirm its capability to extend the context window and improve inference efficiency. Furthermore, ablation studies provide evidence for the effectiveness of the introduced context information injection mechanism.

**Weaknesses:**

**Weakness**

(1) The research motivation is not sufficiently clear. The point raised in the introduction—that "specialized attention patterns may cause incompatibility with high-performance attention implementations"—does not adequately motivate the proposed method. For instance, prompt compression methods (e.g., ICAE and UniICL) can also improve long-context inference efficiency without requiring specialized attention mechanisms. However, the authors neither compare their method with these alternatives nor clarify the uniqueness of the problem their approach aims to solve.

(2) The abstract requires smoother expression. For example, the definition of "self-injection" is repeated, resulting in redundant content. Additionally, the mention of "sender" and "receiver" concepts in the abstract may lead readers to believe they are important modules, yet these terms are only mentioned there, which causes confusion.

(3) If no information has been overlooked, Tables 2 and 3 do not report the memory usage or inference latency of the baseline methods. Without aligning these factors, it is difficult to ensure a fair comparison between different methods.

**Questions:**

**Suggestions**

(1) Improve the expression in the Abstract and Introduction to enhance readability.

(2) Include the memory usage or inference latency of all compared methods in Tables 2 and 3 to ensure a fair comparison.

(3) If feasible, conduct experiments on larger models to validate the scalability of the approach.

---

> ### Author Response · Authors · 2025-11-16
> **Response to reviewer AWmy**
>
> Thank you for the detailed and constructive feedback. Below we provide our point-by-point responses.
>
> ## W1 Motivation
> We carefully read the two referenced works. While they apply prompt compression technique to enhance efficiency---their practical assumptions, experimental settings, and targeted capabilities differ substantially from those addressed by SharedLLM:
>
> - **ICAE fails to show scalability to extremely long inputs**. As shown in Figure 10 of their paper, the evaluated input lengths are typically below 2K tokens, which falls within the default context window of the base models such as Vicuna. In contrast, SharedLLM is trained on 8K inputs and evaluated on sequence up to 1M tokens, demonstrating genuine long-context capability rather than short-range extrapolation.
> - **UniICL is tailored for In-context learning tasks under the well-segmented and highly structured input setting**. UniICL adopts RAG-style pipeline for demonstration selection and integration. In their setting, **the input demonstration is precisely segmented into individual chunk**, resulting in a more structured and constrained compression problem. Conversely, SharedLLM assumes **a generic long sequence with no explicit boundary as input**---a much more challenging and realistic scenario.
> Moreover, SharedLLM is not limited to the in-context learning task. It is designed as a general long-context task solver, covering summarizaiton, document-QA, code problem, etc.
>
> Thus, the motivation of SharedLLM is distinct:
> 1. To extend the context length of existed short-window LLMs, instead of just compression and acceleration;
> 2. To enable the general purpose long-context modeling under unconstrained inputs and diverse tasks, which existing compression-based methods do not cover.
>
> ## W2 & Q1 Writing
> Thanks for raising this point! We have revised the manuscript to clarify the ambiguity and improve readability. Please refer to  the updated version for the corrected explanation.
>
> ## W3 & Q2: Efficiency
> Figure 3 reports the time and memory usage across different truncated lengths for the language modeling task. For clarity, we present curves for **representative SOTA methods** in each model categorie: base architecture (base/**YaRN**/LongAlpaca), streaming (SnapKV/OmniKV/**Activation Beacon**), compression (AutoCompressor/**CEPE**), and SharedLLM.
>
> Below we additionally provide the time/memory usage of all these methods on LLaMA-2 reproduced from the source code:
> | Method | Time (ms) | Memory (GB) |
> | ---- | :----: | :----: |
> | Base | 21.99 | 79.63 |
> | StreamingLLM | 21.57 | 63.39 |
> | LongAlpaca | 22.84 | 80.17 |
> | SnapKV | 18.89 | 59.71 |
> | OmniKV | 17.54 | 57.33 |
> | Activation Beacon | 11.65 | **45.21** |
> | SharedLLM | **9.81** | 49.69 |
>
> It can be observed that SharedLLM enjoys the shortest latency, while Activation Beacon minimizes the memory usage at inference time.
>
> ## Q3: Scalability on model size
> We report the results of SharedLLM on LongBench as well as some reproduced baselines with Llama-2-13B.
> | Model | SD-QA | MD-QA | Summ. | FS | Code |
> | ---- | ---- | ---- | ---- | ---- | ---- |
> | Base |  26.74 | 27.98 | 23.66 | 66.50 | 40.88 |
> | SnapKV | 27.36 | 27.84 | 24.02 | 63.96 | 41.74 |
> | Activation Beacon |  31.39 | 35.62 | **28.27** | 70.80 | 50.94 |
> | SharedLLM | **33.72** | **36.84** | 27.56 | **71.18** | **52.36** |
>
> SharedLLM yields superior results compared to other baselines, which validates its scalability to larger size of LLMs.

---

> ### Author Response · Authors · 2025-11-28
> **Gentle reminder**
>
> Dear Reviewer,
>
> We sincerely thank you once again for your valuable time and effort in reviewing our paper and for the thoughtful feedback you have provided. In response, we have prepared a targeted and detailed reply and updated the manuscript accordingly. However, we have not yet received your comments or follow-up questions.
>
> We kindly ask that you take a moment to review the information we have provided and share any additional comments or feedback. This will help us ensure that all concerns are addressed before the discussion period concludes.
>
> Best regards,
> Authors

---

### Author Response · Authors · 2025-11-28
**Gentle Reminder: Please Review Responses**

Dear Reviewers,

As the discussion period is coming to an end, we would like to kindly remind you that our responses are currently awaiting your review in OpenView. Please take a moment to check whether our replies have addressed your major concerns and let us know if further explanation or additional results are required so we can incorporate them before the deadline.

We greatly appreciate your time and great effort during this time again!

Best,
Authors

---

### Author Response · Authors · 2025-11-29
**Rebuttal Summary**

Dear Area Chair,

We thank all reviewers and chairs for their diligent efforts throughout the review and rebuttal process. As we were informed that reviewers will no longer be able to modify their responses or scores due to the data leakage issue, we understand that the discussion period has effectively concluded. Below, we provide a brief summary of our rebuttal process:

## Contributions
This work proposes a new architecture for **context window extrapolation, enabling extended context lengths for LLMs during both training and inference**:

- A new architecture, including model initialization, context compression and integration
- The training/inference paradigm

Different from many previous works that focus only on **efficient inference on long-context**, which simply adjusts the inference procedure based on an existed long-context LLM, we aim at creating **a new long-context LLM from short-context LLMs that can perform tasks when input lengths exceed the upper limit that the original short-context LLM cannot handle**---this is the most significant and intrinsic difference from compression-based inference-time acceleration.

## Rebuttal Overview
All reviewers provide constructive suggestions to improve the writing and presentation, as well as discuss to resolve potential confusions in understanding this paper. Below we briefly summarize the key points among the .

### Key Points
- Reviewer AWmy
  - Revise abstract and introduction to provide **clearer motivation, clarification and description** of the methodology.
  - **Memory usage** when performing long-context understanding tasks.
  - Results on **larger-sized LLMs (13B)**.

- Reviewer 4Mtb
  - **Sensitivity to hyperparameters** and **task generalizability**
  - **Details of SharedLLM's internal mechanism**, such as positional embedding and tuning methods.

- Reviewer xDgb
  - **Comparison with system-level methods**
  - **Advantages over test-time methods** (SnapKV, KVzip)

- Reviewer CrBs
  - Performance on **longer input**
  - Qualitative rexanples to show the **split-and-select** process

For each point of concern, we provide a targeted response supported by both descriptive explanations and quantitative evidence. In addition, we have made the corresponding revisions as follows:

### Revision of Submission
- **Revise abstract and introduction** , as per the suggestion from (Reviewer) AWmy.
- **Add related works and comparison** in Section 1 and Section 4, as per the suggestions from AWmy and xDgb.
- **State the purpose of benchmarks**, i.e., which ability to reflect in section 3.2., in response to 4Mtb
- **Add qualitative sensitivity analysis** on hyperparameters in section 3.4, in response to 4Mtb
- **Provide a living example** (in both figure and text) showing how SharedLLM processes NIAH in Appendix D.

----
## Reviewer's Feedback

Two reviewers sent feedbacks, and we actively handle their concerns:
- **Reviewer xDgb** expressed satisfaction with our replies and proposed a new follow-up questions, and **updated the score from weak reject to weak accept**. For his follow-up questions, we provided new evidence, which we believe sufficiently addresses the remaining concerns, but unfortunately cannot receive his feedback as the response function has been closed due to the leakage issue.
- **Reviewer 4Mtb** acknowledged our response and maintain his/her score at the high level. We follow his/her suggestions to incorporate the key points into the updated version of the paper.


Regretably, we fail to receive further feedback on our from **Reviewer AWmy** and **Reviewer CrBs** till the last moment---even after we requested assistance from the area chair. We made every effort to clarify their concerns by preparing further explanations and futher explanation and new experimental results, and we have committed to incorporating the supplementary content into the final version of the paper.

----

We hope this summary helps provide a clear overview of our rebuttal process and assists you in making the final decision.

Best,
Authors

---

### Meta-Review · Area_Chair_KpU2 · 2026-01-05

**Summary:**

1. The reviewers have concerns about the clearness of abstract and the motivation of the introduction. (Reviewer AWmy)
2. The reviewers have concerns about the comparison with related works (Reviewer AWmy, xDgb)
3. The reviewers have concerns about evaluation fairness and evaluating on the long sequence length scenario (Reviewer 4Mtb, CrBs)

**Reviewer Concerns:**

1. I've re-read the revised abstract. The revised abstract clearly presented the motivation and the approach. I think the abstract concern is addressed. For the motivation concern, I checked the rebuttal, the author explained the difference b/w the proposed approach and the mentioned references. The introduction is also updated with that. I think this is addressed.
2. For the comparison with the related works, the reviewer xDgb agreed the concern is addressed.
3. For the concerns about the evaluation fairness, the reviewer 4Mtb resolved that after the rebuttal. Author also provides enough evidences for addressing the concerns raised by Reviewer CrBs for the long sequence length.

**Reviewer Scores:**

All the reviewers agreed with acceptance except Reviewer AWmy. By checking the review and the rebuttal carefully, I agree that the rebuttal shall address the concerns raised by the reviewer. The rejection proposed by the reviewer shall be improved.

---

### Decision · Program_Chairs · 2026-01-26

Accept (Poster)